# Seasonal to Multi-Decadal Shoreline Change on a Reef-Fringed Beach

**Thibault Laigre** [1,2,*], **Yann Balouin** [3], **Deborah Villarroel-Lamb** [2] **and Ywenn De La Torre** [1]

1   Bureau de Recherches Géologiques et Minières, Parc d'Activités Colin—La Lézarde, 97170 Petit Bourg, France; y.delatorre@brgm.fr

2   Department of Civil and Environmental Engineering, St. Augustine Campus, The University of the West Indies, St. Augustine 999183, Trinidad and Tobago; deborah.villarroel-lamb@sta.uwi.edu

3   Bureau de Recherches Géologiques et Minières, Université de Montpellier, 1039 Rue de Pinville, 34000 Montpellier, France; y.balouin@brgm.fr

*   Correspondence: t.laigre@brgm.fr

**Abstract:** This study investigates the shoreline dynamics of a Caribbean reef-lined beach by utilizing a long-term satellite dataset spanning 75 years and a short-term, high-frequency dataset captured by a fixed camera over 3 years. An array of statistical methods, including ARIMA models, are employed to examine the impact of storms and potential cyclical influences on the shoreline dynamics. The findings indicate that significant storm events trigger a substantial retreat of the vegetation limit, followed by a slow recovery. Given the current frequency of such major events, complete recovery may take several decades, resulting in a minor influence of cyclones on the long-term erosion trend, which remains moderate. The short-term shoreline evolution is primarily driven by the annual cyclicity of the still water level, which generates an annual oscillation—an insight not previously reported. In the context of climate change, alterations to sea-level rise and cyclone frequency could disrupt the observed dynamic equilibrium at different timescales. Such changes could result in an alteration of existing cyclicities, disturbance of recovery periods, increased long-term shoreline retreat rates, and potentially affect overall coastal resilience over time.

**Keywords:** shoreline monitoring; remote sensing; satellite imagery; video imagery; coral reef; upperbeach vegetation; Caribbean

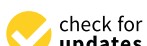



## 1. Introduction

Sandy beaches dynamically adapt to the wave climate, resulting in coastline retreat or expansion. This change occurs both seasonally and interannually, a phenomenon well-documented in the literature [1–9]. Provided there is no significant erosion or accretion trend, the beach's stability maintains a dynamic balance within a certain range of wave conditions [10]. However, severe storms can disrupt this balance, inducing extraordinary transformations in beach morphology [11–13]. Following these changes, the beach morphology may either revert back to its pre-storm state under smaller, post-storm waves [14,15], or the changes might become permanent [16]. In reef-lined beaches, up to 98% of incoming wave energy can be dissipated by the reef system, attributed to the system's shallowness and intricate bottom structure [17–20]. Even if less attention has been received in the literature, similar dynamics concerning the impact of major storms have been observed in existing studies, with patterns of significant erosion followed by emerging recovery periods [21–23]. However, a knowledge gap persists concerning the interannual and seasonal dynamics of these reef-lined beaches. This study focuses on a reef-lined beach in the Caribbean, a region often exposed to vigorous hydrodynamic events such as North Atlantic winter swells [24] and cyclones [25]. These latter generate extreme waves, with significant wave heights ($H_s$) reaching up to 12 m [26]. Past research in the Caribbean has examined the erosional effects of tropical cyclones on coastlines. A decade-long study on the smaller

islands of the Eastern Caribbean found that 70% of beaches underwent significant erosion, with rates three times higher on hurricane-exposed shores than the usual background rate [27]. As a result, understanding coastal processes affecting shoreline evolution in the context of reef-lined beaches at different timescales is crucial. Direct shoreline monitoring is challenging and resource-intensive. Remote sensing technologies are a good alternative to direct observations, as the use of georectified images from aerial, airborne, and Unmanned Aerial Vehicle (UAV) sources [14,28], as well as fixed-video systems [29–31] may be used to study shoreline evolution over time. For this study, two image datasets were used to monitor shoreline evolution: a historical collection of aerial and satellite images dating back to the 1950s, and a more recent fixed-video system dataset that captured high-frequency shoreline changes over three years. Although invaluable, such high-frequency, in situ shoreline observations over extended periods are rarely available in reef environments. Remote shoreline monitoring is common [4,32–34], and some preliminary studies have been conducted in fringed reef environments [35]. However, to our knowledge, no study has yet provided a continuous multi-year dataset that allows analysis of shoreline variability at daily to multi-annual scales on a reef-lined beach. Therefore, the present study offers a unique insight into the complex interplay of factors influencing shoreline behavior in a reef-fringed pocket beach setting. We analyzed the variables affecting the shoreline over extended periods and linked them to short-term processes. This integrated approach will enable a comprehensive understanding of the intricate processes affecting shoreline dynamics in such environments. This research centers around two objectives: (a) to investigate multi-decadal shoreline variability and the effect of cyclones on these dynamics; (b) to explore the impact of various variables, such as sea-level variations or storm-induced waves, on the short-term response of the shoreline.

## 2. Study Area and Methodology

### 2.1. Study Site Description

Located within the Lesser Antilles on Guadeloupe Island, France, lies the small beach of Anse Maurice (Figure 1a,b). This stretch of coastline, approximately 200 m long and between 5 and 20 m wide, has been subject to consistent erosion, resulting in an average retreat of 20 m from 1950 to 2013, as highlighted by [36]. Following this study, the main cause of erosion could be attributed to sand mining and the deepening of the coral reef, which is associated with degradation and the resulting increase in remaining wave energy on the reef flat [37]. These potential causes are regional, and no hypotheses specific to the site were proposed. The beach is surrounded by a fringing reef primarily consisting of dead *Acropora palmata* colonies draped in algae. While the reef has lost some of its structure, it still displays several meter-high complexes. The reef flat is characterized by discontinuous coral structures, each about 1 m high and several meters wide, interspersed with small patches of living branching and encrusting corals. Notably, the southern region of the site hosts a discernible channel where coral structures are less dense and extend deeper.

Vegetation on the upper beach constitutes an ecological succession, progressing from crawling species to shrubs and finally to trees. The most widespread species include *Ipomoea pes-caprae*, which forms the frontline, followed by *Coccoloba uvifera* shrubs, and a few non-native coconut trees. However, human activity, specifically trampling and goat grazing, has significantly impacted the vegetation at Anse Maurice. The lower vegetation layers suffer the most, impeding the growth of crawling plants and seedlings, and consequently disrupting the development of a new tree generation. Erosion becomes a more pressing issue as the soil loses its protective cover.

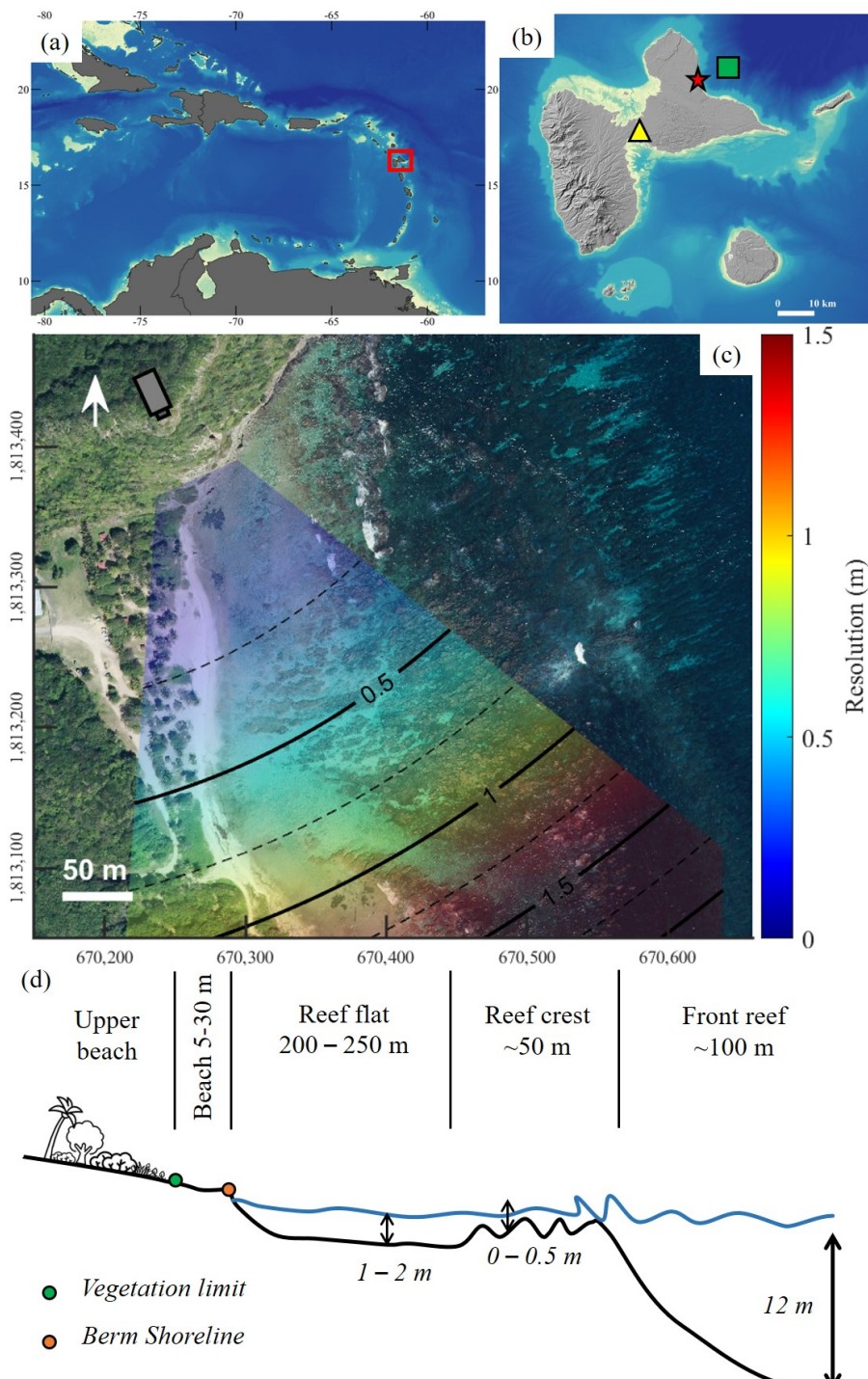

**Figure 1.** (**a**) Location of Guadeloupe Island in the Caribbean and (**b**) the Anse Maurice beach (red star) on the East coast of the Island. The green square indicates the point of extraction of the MARC model waves and the yellow triangle indicates the tide gauge. (**c**) Anse Maurice beach orthophotography with the imagery system field of view and position, system coordinates: RGAF09 (EPSG: 5490) The color indicates the resolution of the data as a function of the position. (**d**) Typical cross-shore profile at the Anse Maurice site with characteristic areas and position of the vegetation limit and berm shoreline.

Anse Maurice is situated on a wave-dominated coast with offshore swells showing significant annual variation, averaging around 1.2 m. The site is open to potent Atlantic

swells, predominantly during the winter months (December to March), originating from the north to east-north-east directions [24]. In addition, cyclonic events between July and November can generate the most potent waves that can exceed 10 m in $H_s$. However, only swells from the north to east direction directly impact the site due to its orientation. Apart from these two high-energy wave regimes, trade wind-generated waves affect the area throughout the year, with $H_s$ fluctuating between 0.5 and 2 m [38,39]. The most significant flooding risk comes with the occurrence of tropical cyclones [40]. The site experiences a semidiurnal microtidal range with diurnal and mixed inequalities, with a mean range of 0.25 m, fluctuating between 0.1 m during neap tides and 0.6 m during spring tides [41]. The still water level ($\eta_0$) is also influenced by non-tidal forcings. At the event level, atmospheric pressure and wind can contribute significantly (i.e., storm surge), while at the seasonal level, forcings such as atmospheric pressure and steric expansion cycles may affect the sea level. Annual fluctuations in atmospheric pressure in the Caribbean are linked to the Inter-Tropical Convergence Zone's (ITCZ) movement [42].

### 2.2. Hydrodynamic Conditions

#### 2.2.1. Waves and Still Water Level

This research employed both $\eta_0$ data and offshore wave conditions for two key reasons: (1), to ensure the selection of images under similar hydrodynamic circumstances for the detection of shoreline changes, and (2), to examine the impact of these variables on shoreline alterations. The data for the $\eta_0$ was acquired from the Pointe-à-Pitre tide gauge, which is nearest to the study location. The French Hydrographic Service (SHOM) is responsible for these data (which can be found at [43]). The astronomical tide was extracted from the TPXO9-atlas (1/30° resolution) database [44], and anomalies were computed by determining the difference between the global $\eta_0$ and TPXO tides. Offshore wave conditions were derived from the MARC model's outputs. The MARC model is a regional scale reanalysis of the WAVEWATCH III® model, with simulation results provided in real-time by the IFREMER (French Institute for Sea Exploitation Research) at [45]. Hydrodynamic variables were condensed to daily values as higher frequencies are incompatible with shoreline changes. $\eta_0$, over this timescale, mainly fluctuates due to non-astronomical variability [46]. As a result, the anomaly of daily maximum $\eta_0$, i.e., the deviation from the astronomical tide, was denoted as $\Delta d\eta_0$. In addition, for each day, the cumulative wave power in kWh/m was estimated following the method of [47]. This procedure provides data on the significant wave height ($H_s$) and peak period ($T_p$) as a single variable, merging the temporal and hydrodynamic dimensions. It is utilized to examine the wave power variation throughout the year. It is defined as follows:

$$d\Sigma P = \int_{h1}^{h24} P\Delta t \tag{1}$$

where $h1$ and $h24$ are, respectively, the first and the last data extraction of the day and $P$ is the wave power per meter of wavefront length in kW/m, defined as:

$$P = \frac{\rho g^2}{64\pi} H_s T_p \tag{2}$$

where $\rho$ is the water density and $g$ is the acceleration due to gravity.

#### 2.2.2. Cyclones Tracks

The analysis of potential cyclones impacting the Anse Maurice beach employed the HURDAT2 dataset [48,49]. It is essential to consider actual hurricane tracks, as numerical models utilized for offshore wave condition extraction may not accurately replicate cyclonic waves due to the speed, variability, and unpredictability of these events [50–52]. The offshore wave dataset has only been available for about a decade, and no wave data exists for the historic period. Therefore, this hurricane track dataset serves as a valuable resource for interpreting shoreline changes. The exposure area is defined based on the location and

geographical layout of Anse Maurice within the Guadeloupe archipelago. It was assumed that any cyclone traversing this area could potentially generate waves reaching the shore. The exposure area is defined as a circular arc spanning 900 km in radius, extending from 3° to 93°. Its boundaries encompass the eastern coast of Grande-Terre in the north (culminating at the Grande-Vigie headland) and the Pointe-des-Châteaux headland and Désirade Island to the south. The 900 km arc limit was chosen due to the observed fact that no event beyond this distance has exhibited a significant effect at the site. The most impactful distant event noted was Hurricane Teddy. The eye of this Category 5 hurricane passed within 850 km of the study area at its closest point. An intensity threshold was also set based on the sustained wind speed within the exposure area. This threshold served to filter out cyclones from the database that were deemed less likely to significantly impact the shoreline. In the historical analysis spanning from 1947 to 2022, only events that demonstrated a sustained wind speed exceeding 116 km/h while within the exposure area were taken into account. This limit effectively separates tropical storms from hurricanes [53,54], and it aids in filtering out events of milder intensity that are unlikely to result in observable shoreline alterations. Therefore, both the exposure area and the intensity threshold were utilized to identify the events most likely to significantly impact the shoreline.

### 2.3. Shoreline Datasets

### 2.3.1. Airborne, Satellite and UAV Derived Dataset

A collection of aerial and satellite images from the IGN (the French National Institute for Geographic and Forestry Information) and the CNES (the French Center of Spatial Studies) was used to assess shoreline changes at the study site over the historical period. The collection includes black-and-white historical aerial photographs from the IGN, Pleiades, and SPOT satellite images, and images from UAV surveys. A total of 24 shorelines were extracted from 1947 to 2022 (See Table 1).

Rectified satellite imagery was utilized for the study. UAV images were processed through the Agisoft Metashape software, with ground control points collected via real-time kinematic global navigation satellite systems (RTK-GNSS) employed to create orthophotographies [55]. Images acquired from aircraft surveys were rectified using the Qgis geospatial data abstraction library (GDAL) Georeferencer. This process was grounded on identifiable features shared between a calibrated image and the raw image sourced from the aircraft. Given that this latter method of rectification is not as precise, due to the image resolution and quality (such as sun and shades, and black and white images), we have taken into account a reasonable margin of error of 5 m for the shoreline extraction process. This value is based on the results of studies using similar resolution images [56,57]. Subsequently, the position of the shoreline was digitized for each orthophotography using Qgis.

**Table 1.** Details of the images used for shoreline detection for the historic period, including date taken, source, type of photography, capturing device, and image resolution. "??" is indicated when the information about the day or the month where the picture was taken is unknown.

| Id | Date | Source | Photography Type | Device | Resolution (m) |
|---|---|---|---|---|---|
| 1 | 25 February 1947 | IGN | Argentic | Aircraft | 1 |
| 2 | 25 August 1948 | IGN | Argentic | Aircraft | 0.4 |
| 3 | 20 December 1950 | IGN | Argentic | Aircraft | 0.9 |
| 4 | 19 March 1954 | IGN | Argentic | Aircraft | 0.2 |
| 5 | 9 December 1955 | IGN | Argentic | Aircraft | 0.4 |
| 6 | 22 March 1964 | IGN | Argentic | Aircraft | 0.7 |
| 7 | 6 February 1969 | IGN | Argentic | Aircraft | 0.7 |
| 8 | 3 April 1975 | IGN | Argentic | Aircraft | 0.4 |
| 9 | ?? March 1979 | IGN | Argentic | Aircraft | 0.5 |
| 10 | 1 January 1980 | IGN | Argentic | Aircraft | 0.5 |

**Table 1.** *Cont.*

| Id | Date | Source | Photography Type | Device | Resolution (m) |
|----|------|--------|------------------|--------|----------------|
| 11 | 20 February 1984 | IGN | Argentic | Aircraft | 0.4 |
| 12 | 16 March 1988 | IGN | Argentic | Aircraft | 0.4 |
| 13 | 14 October 1989 | IGN | Argentic | Aircraft | 0.2 |
| 14 | 1 February 1999 | IGN | Argentic | Aircraft | 0.6 |
| 15 | ?? ?? 2004 | IGN | Numeric | Aircraft | 0.5 |
| 16 | 21 February 2010 | IGN | Numeric | Aircraft | 0.3 |
| 17 | ?? ?? 2014 | CNES\IGN | Numeric | Pleiades satellite | 0.5 |
| 18 | ?? ?? 2015 | CNES\IGN | Numeric | Spot 6-7 satellite | 2 |
| 19 | ?? ?? 2016 | CNES\IGN | Numeric | Spot 6-7 satellite | 2 |
| 20 | ?? ?? 2017 | IGN | Numeric | Aircraft | 0.2 |
| 21 | ?? ?? 2018 | CNES\IGN | Numeric | Pleiades satellite | 0.5 |
| 22 | ?? ?? 2019 | CNES\IGN | Numeric | Pleiades satellite | 0.5 |
| 23 | 28 September 2020 | BRGM | Numeric | UAV | 0.1 |
| 24 | 21 October 2022 | BRGM | Numeric | UAV | 0.1 |

### 2.3.2. Fixed Video System Derived Dataset

The fixed-camera was installed in April 2019 and remained functional until December 2022, the period within which data for this research were collected. To provide the data on a real-world scale, as we did for UAV, aircraft, and satellite imagery, the device was calibrated using the method outlined by [58], using ground control points captured with an RTK-GNSS system. This rectification process facilitated the conversion of image coordinates (U, V) into world coordinates (X, Y, and Z), thereby enabling the quantitative analysis of coastal state indicators. With respect to resolution, our installed device ranged from 0.03 m at the north of the beach, the closest position to the camera, to 0.9 m at the southern end (as illustrated in Figure 1c). It is worth noting that the resolution decreases markedly towards the south, reaching several meters beyond the area of interest. Given that hydrodynamic conditions, such as wave activity and $\eta_0$, can significantly influence shoreline appearance, it is vital to extract and compare images that share similar conditions. Without this standardization, changes in the shoreline could be erroneously attributed to morphological shifts, rather than the more likely influences of water level fluctuations or wave setup. As such, a filter was applied to both $H_s$ and $\eta_0$ to select images that exhibited comparable levels of agitation and $\eta_0$. To establish the $\eta_0$ filter, the mean $\eta_0$ value over the duration of the study was first computed. Subsequently, an acceptable tolerance of plus or minus 0.1 m was selected around this mean value. On a daily basis, the filter sought out the image whose $\eta_0$ value was closest to the mean, but still within the acceptable tolerance. In cases where no such value was found within the buffer, no image was selected for that day. Furthermore, images captured under two specific conditions were excluded: those with $H_s$ values below 0.7 and those above 2 m. Rare instances of $H_s$ values falling under 0.7 m could potentially overstate the offshore positioning of the shoreline, while $H_s$ values exceeding 2 m, indicative of agitated conditions, could result in the erroneous estimation of an overly onshore shoreline. The positions of the shoreline were then digitized using a specially developed MATLAB tool, which enabled us to analyze and compare data with precision and consistency.

### 2.3.3. Shoreline Digitalization and Processing

To digitize the shoreline, we relied on two key markers (see Figure 1d):

- The vegetation limit: This represents the boundary between the last vegetated area and the sandy beach. Serving as an indicator of the beach's mid to long-term evolution (ranging from monthly to multi-annual scales), the vegetation boundary is typically affected only by storm events.
- The berm: This is identifiable either as a sedimentary vertical bulge on the beach or through marine deposits. The berm offers a more dynamic perspective, which informs

the beach's evolution, from short to long term (from weekly to multi-annual scales) changes.

Although these two markers provide different insights, their combined use offers a more comprehensive understanding of the beach's overall evolution over time. However, for the images taken by the fixed camera system, only the berm shoreline was digitized, because the vegetation limit could not be extracted from these images for the entire beach. For each shoreline dataset, analyses were performed using a similar approach to the Digital Shoreline Analysis System (DSAS), a widely used ArcGIS extension for assessing shoreline position changes [59]. This involved generating transects every 5 m perpendicular to the shoreline position, as seen in Figure 2. The intersection of each transect with the shoreline was extracted and used for further analysis.

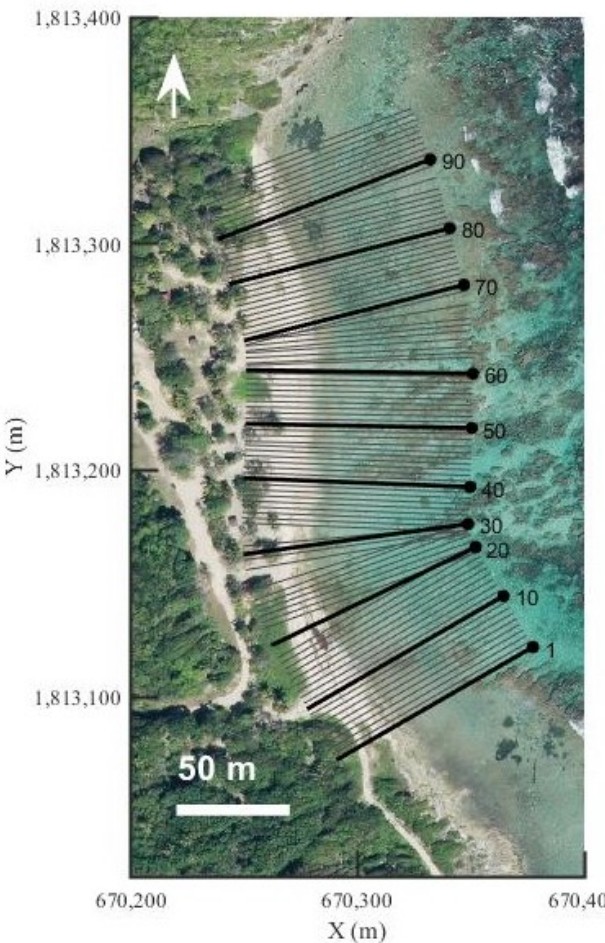

**Figure 2.** Transect (grey and black lines) used to assess shoreline changes, system coordinates: RGAF09 (EPSG: 5490).

On each transect location, several key metrics were calculated, including the position with the maximum retreat or advance, the rate of evolution, the net change, and the deviation from the mean position on the transects as well as for the whole shoreline, as in [1] or [4]. This deviation was computed using the following formula:

$$\Delta S_{i,t} = S_{i,t} - \frac{\sum_{t=1}^{N} S_{i,t}}{N} \tag{3}$$

In this formula, $\Delta S_{i,t}$ stands for the alongshore position of transect $i$ at time $t$, $\frac{\sum_{t=1}^{N} x_{i,t}}{N}$ is the sum of the alongshore positions of transect i across all time points, and N is the total number of shorelines. Then, the derivative of $\Delta S_{i,t}$ was calculated to evaluate the

temporal trends of shoreline change (i.e., whether the shoreline was experiencing accretion or erosion). This was done using the following formula:

$$\Delta S'_{i,t} = \frac{\partial}{\partial t} S_{i,t} - \frac{1}{N} \sum_{t=1}^{N} \frac{\partial}{\partial t} S_{i,t} \tag{4}$$

In this formula, $\delta/\delta t$ represents the derivative with respect to time, indicating how the associated variable changes over time. Values of $\frac{\partial}{\partial t} \Delta S_{i,t}$ that are positive suggest a tendency towards accretion, while negative values imply a trend towards erosion. Furthermore, to evaluate shoreline change across the entire beach, the alongshore average, which is the spatially averaged shoreline position at time t, was compared with the global average position of all shorelines, represented as $\overline{S_t}$. The calculation is expressed as follows:

$$\overline{S_t} = x_t - \overline{t} \tag{5}$$

where

$$\overline{x} = \frac{\sum_{i=1}^{N} x_i}{N} \tag{6}$$

In this context, $xt$ refers to the average alongshore position of a specific shoreline (shoreline at time $t$), while $\overline{x}$ represents the mean of all the alongshore positions across all shorelines. The sum runs over all transects (from $i = 1$ to $i = N$), and the result is divided by the total number of transects ($N$) to get the average. In addition to the aforementioned metrics for the satellite-derived dataset, the mean beach width was also calculated. This is defined as the average of the distances between the vegetation limit and the berm shoreline across all transects. This can be expressed as:

$$\overline{BW} = \frac{\sum_{i=1}^{N}(V_i - B_i)}{N} \tag{7}$$

Here, $\overline{BW}$ represents the mean beach width. $V_i$ and $B_i$ represent the positions of the vegetation limit and the berm shoreline, respectively, on transect $i$. These variables, ($\overline{S_t}$ and $\overline{BW}$), aid in summarizing and evaluating temporal evolution. However, they provide the most accurate results only when shoreline evolution is consistent across the entire beach system, and there is minimal longitudinal variability. Therefore, the usefulness of this approach may vary based on the inherent dynamics of the beach system under study.

*2.4. Data Analysis*

2.4.1. Interrupted Time Series Analysis and ARIMA Model

Several analyses were applied to the datasets in order to evaluate potential causalities and dependencies. Due to the differences in time frequency, distinct processes were applied for long- and short-term analyses. For the long-term analysis, an interrupted time series analysis (ITSA) was used. ITSA is a statistical method employed to gauge the impact of a single intervention or event on a time series of data. For instance, it can measure the effects of policy changes, natural disasters, marketing campaigns, or any other event presumed to cause structural changes in a time series [60,61]. Mathematically, the time series equation can be expressed as:

$$Y = \beta_0 + \beta_1 T + \beta_2 D + \beta_3 P + \epsilon \tag{8}$$

Here, $Y$ represents the outcome variable, $T$ is a continuous variable indicating the elapsed time from the start of the observation period, and $D$ is a Boolean variable that takes the value 0 for observations collected before the intervention and 1 for those collected after the intervention. $P$ is a continuous variable indicating the elapsed time since the intervention ($P$ is equal to 0 before the intervention), and $\epsilon$ represents a zero-centered Gaussian random error. As for the $\beta$ coefficients ($\beta 0$, $\beta 1$, $\beta 2$, and $\beta 3$), they are explained as follows: $\beta 0$ represents the baseline level of the outcome variable Y at the start of the time

series. In other words, it is the expected value of $Y$ when all other factors ($T$, $D$, $P$) are zero. $\beta1$ indicates the trend or rate of change in the outcome variable $Y$ per unit of time before the intervention. It shows how the outcome variable changes over time, independent of the intervention. $\beta2$ represents the immediate impact of the intervention on the outcome variable. It shows the difference in the level of the outcome variable immediately before and after the intervention. $\beta3$ is the change in trend of the outcome variable after the intervention. It indicates whether the intervention has caused the trend of the outcome to increase or decrease compared to the pre-intervention trend.

Autoregressive Integrated Moving Average (ARIMA) models are particularly suitable for ITSA due to several reasons. ARIMA models can model a variety of time series structures, which includes autoregressive (AR) and moving average (MA) components, as well as a trend component (the "integrated" part of ARIMA). AR and MA models assume data are stationary. However, real-world data often violate this assumption, such as trended or cyclical time series. The integrated (I) part of ARIMA transforms non-stationary time series data into a stationary form by taking differences between observations. This flexibility makes them an ideal choice for modeling complex real-world data [60,62]. The autoregressive component (AR) is described as:

$$Y_t = c + \phi_1 Y_{t-1} + \phi_2 Y_{t-2} + \ldots + \phi_p Y_{t-p} + e_t \tag{9}$$

Here, $Y_t$ is the value of the time series at time $t$, $\phi_1$, ..., $\phi_p$ are the parameters of the model, $e_t$ are the values of the error term (white noise), $c$ is a constant, and $p$ is the order of the model. The moving average component (MA) is described as:

$$Y_t = \mu + e_t + \theta_1 e_{t-1} + \theta_2 e_{t-2} + \ldots + \theta_q e_{t-q} \tag{10}$$

Here, $e_t, e_{t-1}, \ldots, e_{t-q}$ are the values of the error term (white noise), $\mu$ is the expectation of $Y_t$ (often assumed to be 0), and $q$ is the order of the model. If the time series $Y$ is non-stationary, it can be differenced once to yield a new series:

$$\Delta Y_t = Y_t - Y_{t-1} \tag{11}$$

If this differenced series is still non-stationary, it can be differenced again:

$$\Delta^2 Y_t = \Delta Y_t - \Delta Y_{t-1} \tag{12}$$

and so forth. Each differencing step, however, results in the loss of one observation. The following equation:

$$\Phi(B)(1-B)^d Y_t = \Theta(B) Z_t \tag{13}$$

represents the full ARIMA model, where $\Phi(B)$ is the AR operator of order $p$, $(1-B)^d$ represents the $I$ operator of order $d$, and $\Theta(B)$ is the MA operator of order $q$. $Yt$ is the time series, and $Zt$ is a white noise series. Moreover, ARIMA models include parameters that allow the model to specifically account for the intervention. This is done by incorporating a "dummy" variable that changes value at the time of the intervention and potentially an additional variable to represent the cumulative effect of the intervention over time [60].

### 2.4.2. Cyclicity and Correlations

To extract insights from the high-resolution camera-derived shoreline dataset, several statistical methods have been employed, starting with a simple linear correlation and proceeding to more complex techniques, including sinusoidal fitting, autocorrelation, and cross-correlation analyses. The initial step involved performing a linear correlation analysis between the shoreline dataset and the two hydrodynamic variables: $d\Sigma P$ and $\Delta d\eta_0$. This analysis helped quantify the degree of linear dependence between these variables. By computing the Pearson $R$ coefficient, a measure of the strength and direction of their linear relationship was obtained. The insights derived from this simple yet powerful analysis pro-

vided a foundation for subsequent, more complex techniques. Subsequently, a sinusoidal fit was conducted on the daily time series to reveal potential annual cyclicities. This was accomplished by fitting the dataset with a least-squares estimate of a sinusoid featuring a periodicity of one year [63]. To gauge the strength of this cyclicity in the initial datasets, the Pearson *R* coefficient between the dataset and its annual periodicity was calculated. Furthermore, autocorrelation and cross-correlation analyses were performed to discern causal relationships between the shoreline evolution and two particular hydrodynamic variables, $d\Sigma P$ and $\Delta d\eta_0$. Autocorrelation was computed to detect any significant peaks that might correspond to an annual cycle. Autocorrelation analysis plays a crucial role in examining a variable's reliance on its past values within a time series, facilitating the identification of recurring patterns [60]. The autocorrelation function (ACF) is mathematically articulated as:

$$\rho(k) = \frac{\sum (x_t - \mu) \cdot (x_{t+k} - \mu)}{\sigma^2} \tag{14}$$

In this equation, $xt$ signifies the data at time $t$, $xt + k$ denotes the data $k$ time periods later, $\mu$ is the mean of the series, and $\sigma^2$ is the series' variance. An ACF plot will be used to illustrate autocorrelations for various lags ($k$). To further probe the relationship between shoreline evolution and the selected hydrodynamic variables, a cross-correlation analysis was used. This method measures the similarity between two distinct time series as a function of the lag applied to one, providing insights into the causal links between variables [60]. The cross-correlation function (CCF) is mathematically represented as:

$$r_{xy}(k) = \frac{\sum (x_t - \mu_x) \cdot (y_{t+k} - \mu_y)}{\sigma_x \cdot \sigma_y} \tag{15}$$

Here, $xt$ and $yt$ symbolize the data at time $t$ for series $x$ and $y$, respectively, $xt + k$ and $yt + k$ represent the data $k$ time periods later, $\mu x$ and $\mu y$ are the respective means of series $x$ and $y$, and $\sigma x$ and $\sigma y$ are their respective standard deviations. A CCF plot for various lags was generated. Peaks in the CCF plot will suggest a correlation between the two series at that specific lag. These correlations were evaluated for statistical significance, ensuring the validity of the findings. The interpretation of the CCF plot will provide information about the time lag at which the series are correlated. For instance, a peak at lag $k$ means that the x-series from $k$ time periods ago is a good predictor for the current y-series. This could indicate a causal relationship between $x$ and $y$. The values of the ACF and the CCF range from $-1$ to 1. A value of $-1$ indicates a perfect negative (or inverse) correlation, while a value of 1 signifies a perfect positive correlation.

### 3. Results

#### 3.1. Multi Decadal Shoreline Observations

Figure 3 displays all 24 shorelines, and their respective ages are illustrated in the legend. The berm shoreline dataset appears very coherent, while the vegetation limit dataset presents greater variability, with one shoreline located relatively far away from the others.

For both shoreline datasets, the overall trend is one of a moderate retreat. The mean shoreline changes between 1947 and 2022 are $-8$ m on the berm shoreline and $-5$ m on the vegetation shoreline. Excepting on the southern extremity of the beach, which is less eroded, the whole beach exhibits almost the same retreat on the berm shoreline. The evolution of the vegetation limit is more chaotic. Two areas showed significant changes: at the northern area of the site with a retreat that exceeded 20 m, and at the southern area with up to a 20 m retreat. Notwithstanding the areas showing major changes, such as at both extremes of the beach or at the center, the beach evolution is milder, with only a few meters retreat. To further analyse the evolution between 1947 and 2022, several processing techniques were applied on the datasets. Firstly, for both shoreline types, the deviation from the mean was calculated on each transect over time (Figure 4a,b). Additionally, the deviation of each shoreline mean to the overall mean (Figure 4c) and the mean beach width

(Figure 4d) were calculated. In order to highlight those events that could potentially affect the beach, the cyclonic events extracted were plotted on Figure 4a,b. Table 2 compiles the wave characteristics of the events extracted from the MARC wave dataset. A total of 11 events were extracted using the methodology described in Section 2.2.2.

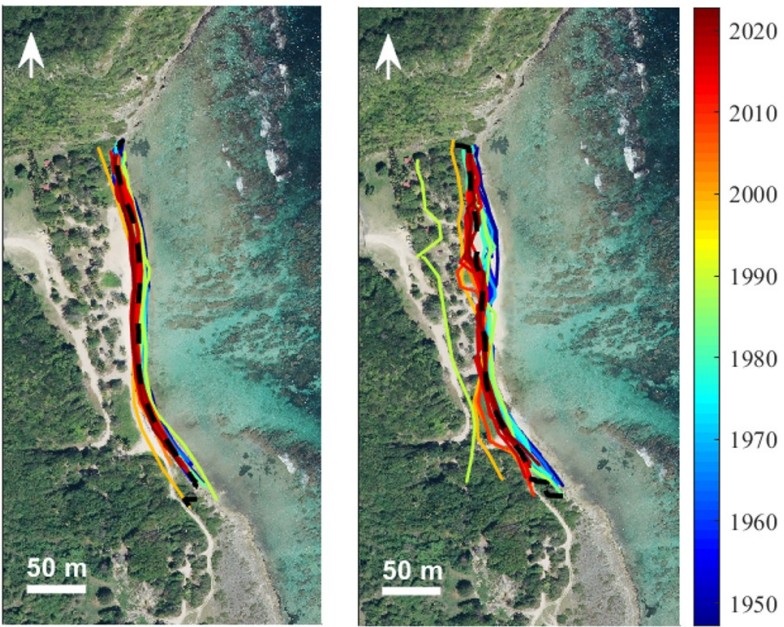

**Figure 3.** Shoreline collection extracted from satellite and airbone datasets, the figure gathers 24 shorelines. The position of the mean shoreline is plotted with a black dotted-line. **Left**: berm shorelines; **Right**: vegetation shorelines (background: 2017 orthophotography from the ©IGN).

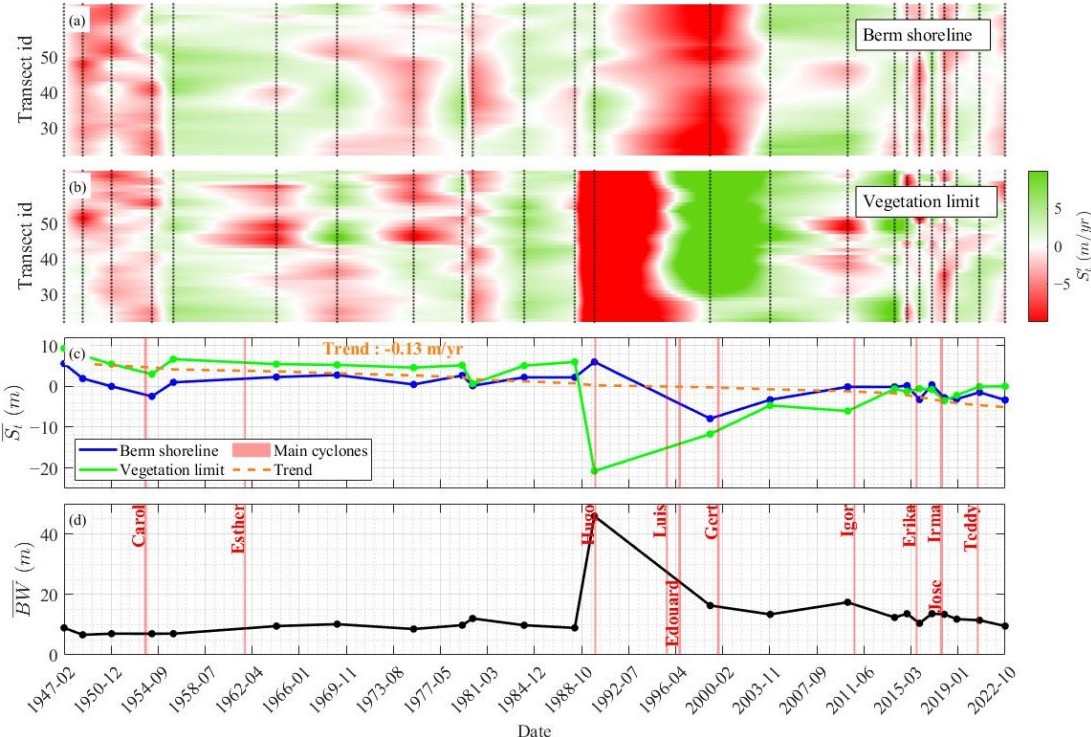

**Figure 4.** Interpolated deviation from the mean by transect of (**a**) the berm shoreline and (**b**) the vegetation shoreline. (**c**) Averaged deviation from the mean (on all transects) for both shorelines. (**d**) Mean beach width.

**Table 2.** Cyclones characteristics for the historic period.

| Event | Date | $H_s$ (m) | $T_p$ (sec) | $D_p$ (°) | Min. Distance (km) |
|-------|------|-----------|-------------|-----------|--------------------|
| Carol | 1 September 1953 | nd | nd | nd | 440 |
| Esther | 1 September 1961 | nd | nd | nd | 830 |
| Hugo | 17 September 1989 | 10 | 7.5 | 90 | 4 |
| Luis | 15 September 1995 | 5.6 | 7.5 | 82.0 | 150 |
| Edouard | 8 August 1996 | 3.6 | 8.6 | 71.0 | 600 |
| Gert | 15 September 1999 | 3.9 | 8.5 | 91.0 | 780 |
| Igor | 1 September 2010 | 4.8 | 8.3 | 62.0 | 740 |
| Erika | 28 August 2015 | 4.5 | 7.0 | 70.0 | 10 |
| Irma | 5 September 2017 | 6.3 | 7.6 | 83.0 | 140 |
| Jose | 9 September 2017 | 4.1 | 9.3 | 78.0 | 190 |
| Teddy | 1 August 2020 | 5.1 | 12.5 | 69.0 | 850 |

The passage of Hurricane Hugo in 1989, known as the most destructive hurricane for Guadeloupe Island in the second half of the 20th century, left a lasting impact on the study site. Hurricane Hugo passed at a few kilometers from the site as a category 4 hurricane. An aircraft image taken one month after the hurricane showed obvious signs of damage, such as widespread sedimentary deposits and destruction of vegetation. The vegetation limit retreated by 25 m, while the berm shoreline advanced by 5 m compared to the last shoreline. Besides Hugo, the dataset shows two significant shifts of the shoreline dynamic occurring after an identified high intensity event. One was associated with the passage of Hurricane Erika in 2015 that passed 10 km north from the site as a tropical storm. The second was observed after the passage of Hurricanes Irma and Jose in 2017 within an interval of a few days. Irma and Jose passed at 140 km and 190 km, respectively, from the site as category 5 and 4 hurricanes. Overall, the site displays a trend of erosion with an average retreat rate of 0.13 m per year.

An ITSA was conducted using an ARIMA model to evaluate the effects of Hugo, Erika, and the combined events of Irma and Jose on the shoreline evolution, and also to determine the duration in order to return to normal conditions (Figure 5). In the case of the berm shoreline dataset, the deviations of the post-Hugo event observations from the predicted trajectory, without the occurrence of the Hugo event, ranged from 9 m immediately after the event to 0 m after a span of 25 years (i.e., in 2014). The Erika event, which occurred in 2015, triggered a retreat of 6 m compared to the prediction without the event. This was followed by the Irma and Jose events in 2017, resulting in a retreat of the berm shoreline by 3.5 m from the last shoreline. This sequence of events was accompanied by an erosional trend up to the end of the dataset. For the dataset with the vegetation limit, a deviation of 20 m was noted between the prediction excluding the Hugo event and the observations immediately post-event. This difference gradually decreased over time, aligning with the event-excluded prediction about 25 years after the event. The two events, Erika and the Irma-Jose cluster, caused less pronounced changes. Post-Erika, a deviation of about 1 m was observed from the prediction without the event, while Irma and Jose resulted in an additional retreat of 2 m compared to the last shoreline position (which had already deviated 1 m from the event-excluded prediction, thus leading to a total deviation of 3 m). The predicted and observed shoreline positions converged four years after the Irma–Jose cluster event in 2021.

This dataset provides insights into the long-term impacts of major storm events on the shoreline dynamics. Yet, its spatial and temporal resolutions limit comprehensive understanding of the dynamics. To explore these dynamics with greater precision, a more recent and higher resolution dataset was used, allowing for the assessment of factors influencing the shoreline over shorter timescales.

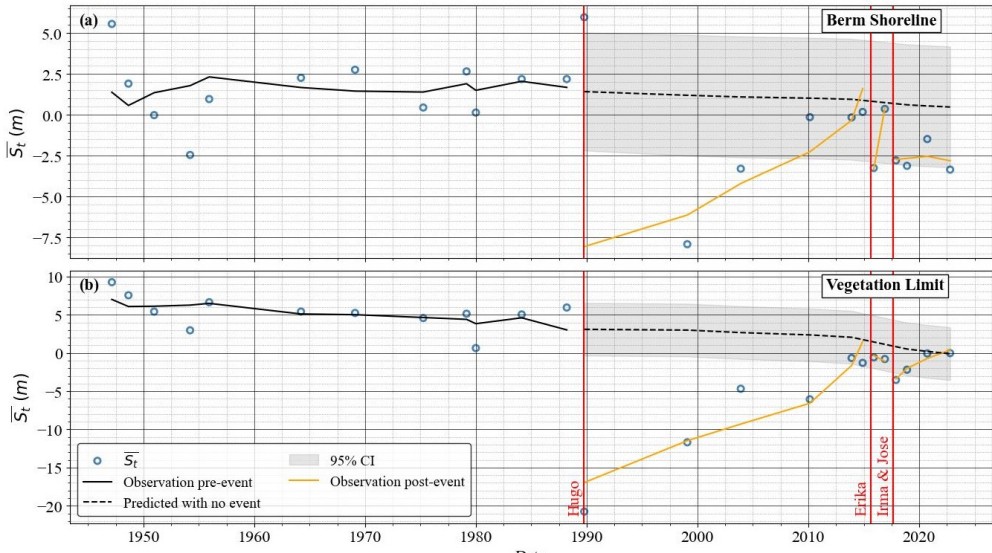

**Figure 5.** ITSA performed using ARIMA model on (**a**) the berm shoreline dataset and (**b**) on the vegetation limit.

### 3.2. High Frequency Shoreline Observations

By applying the hydrodynamics filters followed by an image quality filter on the full image dataset, a total of 60 shorelines were extracted between 13 April 2019 and 13 December 2022, which corresponds to a period of 3 years and 9 months. The initial objective was to extract a shoreline every 15 days. However, a significant number of images were discarded due to the presence of sea spray deposits, water droplets on the lens, or poor lighting conditions. Additionally, periods of camera failure, either complete (camera off for a few days) or partial (weak data transmission), further reduced the number of usable images for shoreline detection. This resulted in a shoreline extraction frequency of approximately one shoreline every 22 days, as shown in Figure 6. The shoreline dataset is presented on a map in Figure 7.

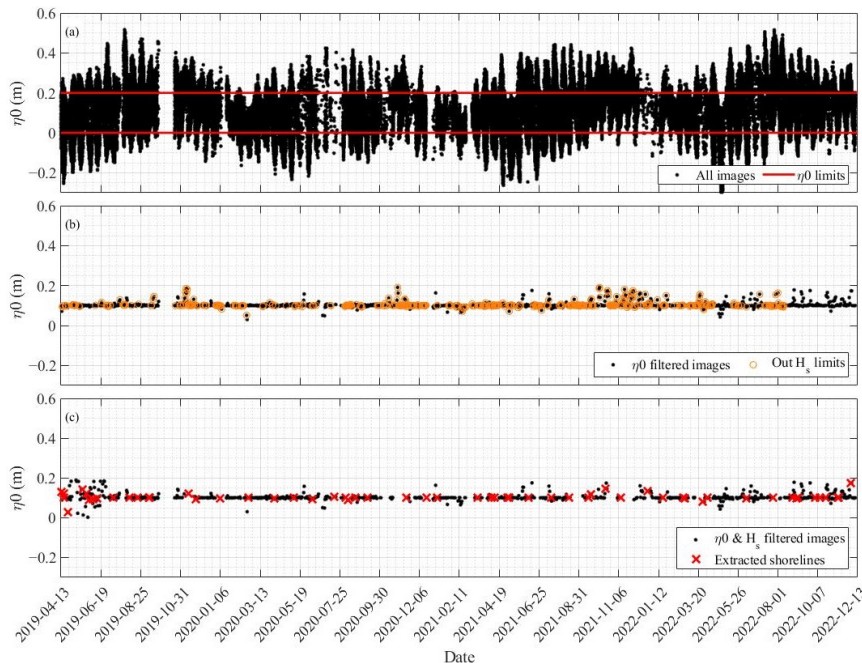

**Figure 6.** (**a**) Corresponding $\eta_0$ for images in the entire dataset. (**b**) Images filtered with the $\eta_0$ filter. (**c**) Images that passed both filter.

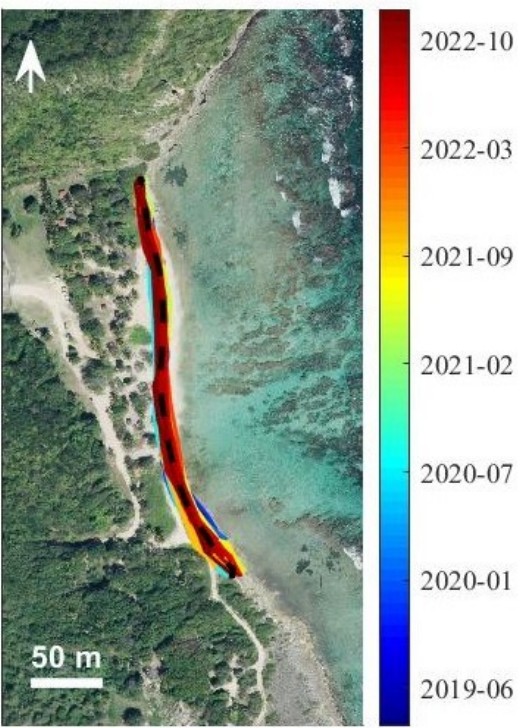

**Figure 7.** Shoreline collection extracted from camera showing the 60 shorelines. The position of the mean shoreline is plotted with a black dotted-line (background: 2017 orthophotography from the ©IGN).

Ten cyclonic events passed through the study area during the recent observation period with the high frequency images from the fixed video system. Their characteristics are presented in the Table 3.

**Table 3.** Cyclones extraction in the study area for the recent period.

| Event | Date | $H_s$ (m) | $T_p$ (sec) | $D_p$ (°) | Min. Distance (km) |
|---|---|---|---|---|---|
| Jerry | 20 September 2019 | 2.6 | 10.3 | 356.0 | 320 |
| Sebastien | 19 November 2019 | 2.0 | 11.2 | 343.0 | 500 |
| Josephine | 15 August 2020 | 2.9 | 9.1 | 40.0 | 300 |
| Laura | 21 August 2020 | 2.9 | 7.7 | 87.0 | 80 |
| Teddy | 18 September 2020 | 5.1 | 12.5 | 69.0 | 850 |
| Grace | 14 August 2021 | 2.7 | 7.4 | 78.0 | 7 |
| Peter | 20 September 2021 | 1.9 | 7.7 | 74.0 | 310 |
| Sam | 29 September 2021 | 2.5 | 9.3 | 85.0 | 610 |
| Earl | 3 September 2022 | 2.0 | 7.6 | 84.0 | 280 |
| Fiona | 16 September 2022 | 4.6 | 7.9 | 98.0 | 0.5 |

As with the satellite-derived dataset, several additional variables were calculated, such as $S_i'$ and $\overline{S_t}$, to assess shoreline evolution. To highlight events that can potentially impact the beach, cyclonic events were identified as outlined in the prior section (Table 3). To analyze the influence of storm events on Anse Maurice, the 10 events extracted were used and incorporated into Figure 8. In addition, the winter events identified through extreme value analysis were also added for context. The annual cycle of the daily hydrodynamic variables, including the differential wave power ($d\Sigma P$) and the anomaly in the maximum daily still water level ($\Delta d\eta_0$), were calculated. It was thought best to use $\Delta d\eta_0$ rather than $d\eta_0$, as it had been identified as having a more significant impact on $\eta_0$ at this timescale [42,46]. Examination of the temporal evolution of transects showed no evident longshore drift pattern emerging. Instead, the system displays a coherent alternation of erosion and

accretion periods (Figure 8a). As no significant longshore variations were observed, the mean evolution of all transects per shoreline provides a reliable representation of shoreline changes. The shoreline variability exhibited a strong annual cycle ($R^2 = 0.70$), characterized by maximum erosion at the end of October, maximum accretion in late April, and an amplitude of 3.73 m relative to the mean position across the entire dataset. Moreover, we observed a trend of accretion at 0.05 m/year (Figure 8b). This trend was determined by comparing a shoreline from June 2019, representing a neutral position in annual variability, with a corresponding shoreline from June 2022. The annual cycle of $\Delta d\eta_0$ also displayed a strong cyclicity ($R^2 = 0.69$), with an amplitude of 0.18 m (Figure 8c). Conversely, the cyclicity of $d\Sigma P$ did not show a good fit to the data ($R^2 = 0.33$) (Figure 8d).

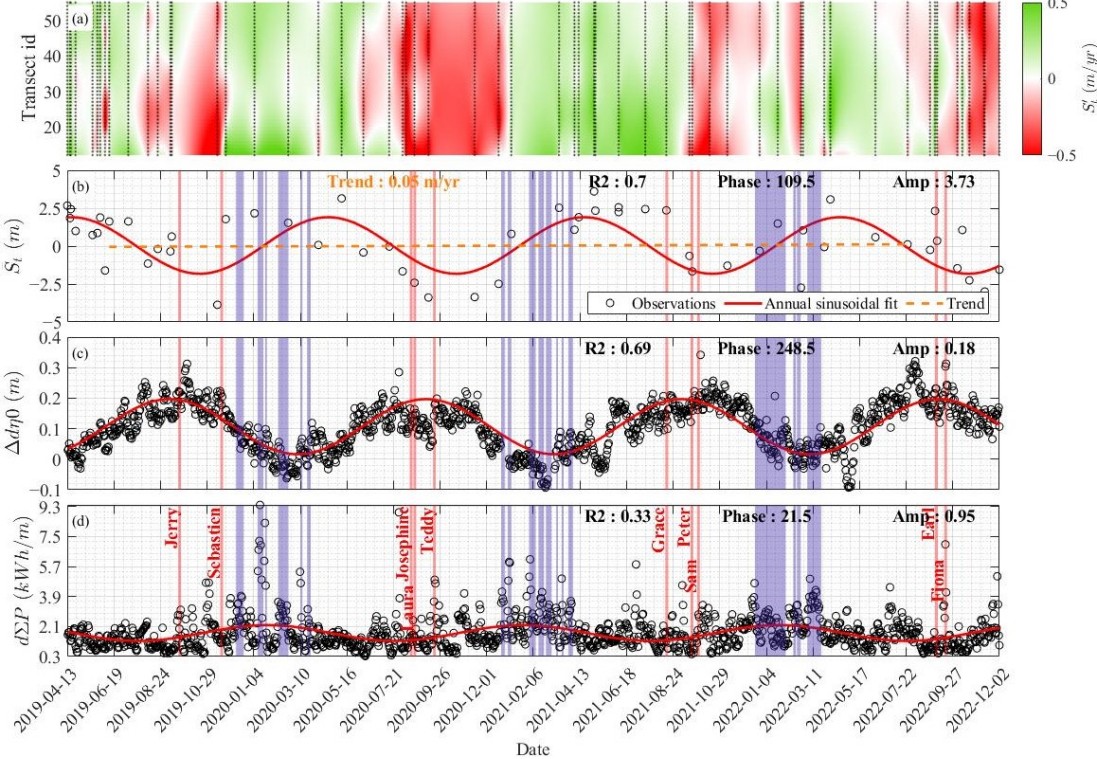

**Figure 8.** (**a**) Interpolated deviation from the mean over each transect, vertical dotted lines represent the exact dates of shoreline extraction and bold vertical dotted lines the dates of DEM extraction by GNSS. (**b**) Averaged shoreline deviation (on all transects) from the mean, (**c**) $\Delta d\eta_0$, and (**d**) $d\Sigma P$. On graphs (**b**–**d**), vertical red lines represent cyclonic events and blue vertical lines represent winter storms.

To assess the dependence of $\overline{S_t}$ on the hydrodynamic variables $d\Sigma P$ and $\Delta d\eta_0$, linear regression was performed (Figure 9). The relationship between $\overline{S_t}$ and $d\Sigma P$ was found to be statistically insignificant, with an R² value close to 0 and a *p*-value >0.05. A weak relationship was observed with $\Delta d\eta_0$ ($R^2 = 0.22$ and *p*-value <0.05). This suggests that if any dependence exists between $\overline{S_t}$ and these hydrodynamic variables, it is likely not linear.

Next, the Autocorrelation Function (ACF) of the three variables were computed (Figure 10a). All variables exhibited an autocorrelation peak at 365 days, implying a strong similarity in the same variable from one year to the next. This is particularly notable for $\overline{S_t}$ and $\Delta d\eta_0$, which demonstrated correlation peaks around one year of 0.45 and 0.35, respectively. However, $d\Sigma P$ only exhibited a peak of 0.16. These findings validate the significant annual cyclicity of $\overline{S_t}$ and $\Delta d\eta_0$ that was identified using the sinusoidal fit. Lastly, the Cross-Correlation Function (CCF) between $\overline{S_t}$ and both $d\Sigma P$ and $\Delta d\eta_0$ were determined (Figure 10b). The CCF with $d\Sigma P$ was low across all lags, remaining under 0.2. Conversely, the CCF with $\Delta d\eta_0$ was substantial, peaking at a lag of −135 days (CCF = 0.4).

This suggests that $\overline{S_t}$ exhibits a robust positive correlation with $\Delta d\eta_0$ when a lag of 135 days is considered. It means that there is a lag of 135 days between the maximum values of $\Delta\eta_0$ and the maximum values of $\overline{S_t}$.

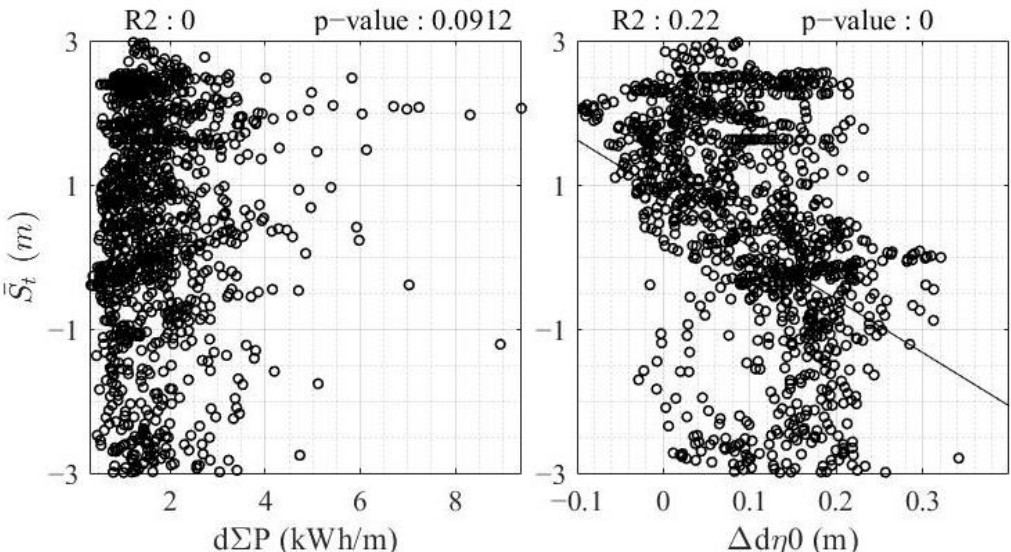

**Figure 9.** Linear regression of $\overline{S_t}$ against (**left**) $d\Sigma P$ and (**right**) $\Delta d\eta_0$.

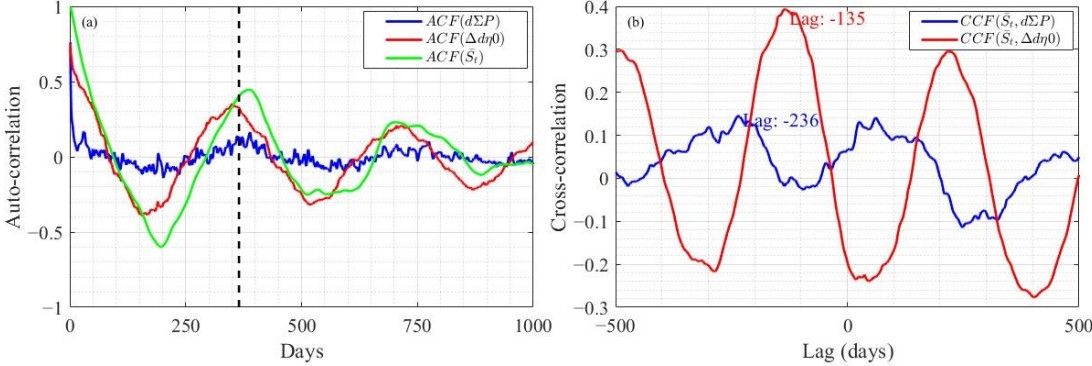

**Figure 10.** (**a**) Autocorrelation Functions (ACF) of variables $\overline{S_t}$, $d\Sigma P$, and $\Delta d\eta_0$. The autocorrelation value corresponding to a year (i.e., 365 days) is represented by a black dotted line. (**b**) Cross-Correlation Function (CCF) of variables $\overline{S_t}$ against $d\Sigma P$ and $\Delta d\eta_0$.

Insights from this analysis underscore a notable time-dependent relationship between $\overline{S_t}$ and $\Delta d\eta_0$. A consistent delay of approximately 135 days is observed in the response of $\overline{S_t}$ to fluctuations in $\Delta d\eta_0$. This lag, calculated to be 139 days based on the sinusoidal fit (derived from the difference of the phase between both annual fits), translates to a temporal offset of roughly 4.5 months. As such, over an annual cycle, the variables demonstrate a relationship that closely approximates a negative correlation (which would be observed with a 6-month offset between maxima). The residual 1.5 months in an approximate negative correlation is indicative of the continued trend in $\overline{S_t}$ post the peak (or trough) of $\Delta d\eta_0$. Specifically, once $\Delta d\eta_0$ reaches its peak, $\overline{S_t}$ persists in its downward trend for an additional 1.5 months. Conversely, when $\Delta d\eta_0$ is at its lowest, $\overline{S_t}$ continues its upward trend for another 1.5 months. The identified trend exhibits an asymmetrical nature, contrary to what might be expected if the offset in maxima were closer to 3 months, which would denote a symmetric phase relationship. This discrepancy indicates that the transitions in $\overline{S_t}$, from increasing to decreasing values, and vice versa, do not correspond to consistent values of $\Delta d\eta_0$ at the corresponding times. This means that for equivalent levels of $\Delta d\eta_0$,

the tendencies in $\overline{S_t}$ display distinct behaviors depending on whether it is on an increasing or decreasing trend.

## 4. Discussion

This study illuminates the mechanisms that influence shoreline dynamics across different timescales on a reef-lined pocket beach. It underscores the importance of high-resolution, long-term data for advancing our understanding of shoreline dynamics and the impacts of major storm events. Three main drivers of shoreline evolution were identified at three different timescales: the scale of a major event (e.g., Hurricane Hugo), the seasonal timescale, and the long-term timescale.

### 4.1. Short-Term Shoreline Evolution: Drivers and Processes

The analysis of the dataset used in this study allowed the discernment of two main processes that act on the shoreline. Extreme storms have induced significant effects lasting up to 20 years on the retreat of vegetation due to high runup and energy dissipation. Interestingly, these storms have little effect on the berm shoreline because of beach flattening and sediment transfer from the upper beach to the beach and reef flat, as cited in previous studies [10,64,65]. On the other hand, less intense storms induced a mild retreat of the vegetation limit and significant erosion on the berm shoreline, with a recovery period of 1–2 years. These less intense storms may also play a role in the reconstruction of the upper beach sedimentary stock [21–23] by transportation of sediment from the nearshore area to the beach and upperbeach by swash processes [66].

The seasonality of the shoreline is forced by the annual cycle in eta0, which impacts the depth over the reef and thus the wave filtering over the year [67–70]. This plays a significant role in the wave energy and runup reaching the shoreline [46], manifesting in a global cycle of ±4 m. Notably, a lag exists between the peak of eta0 and the peak of erosion, and the cycle is not symmetric. This may be attributed to a nonlinear relationship or the involvement of other parameters that imply a threshold. This aspect requires further investigation.

While the response to different storms is quite well-described for open sandy beaches, it has not been extensively studied in the context of reef-lined beaches [21,22]. On the other hand, the influence of eta0 cyclicity on the shoreline is a novel observation. In most cases, seasonality is attributed to wave intensity and/or direction variations over the year [1,2,4,5]. The observations on the historical period were limited by the heterogeneous resolution in time of the images. Nowadays, new techniques are used to monitor shoreline evolution (satellite, video, etc.) with better resolution in time and space, and future observation will provide a better resolution around the passage of events, allowing further interpretation and quantification on the impact on the vegetation limit and berm shoreline as well as on recovery time.

The recent observations obtained using a fixed camera could be complemented with direct measurements, such as regular beach profile measurements utilizing a GNSS-RTK device. Conducting further research to accurately quantify the volume of sediment transfer would be beneficial. This research could also evaluate whether the change occurs at a constant volume or if a portion of the sediment is lost, potentially explaining a long-term erosion trend.

### 4.2. Long-Term Shoreline Evolution

The accumulated residual effect of the aforementioned processes may be the cause of the long-term evolution of the shoreline [71]. The time required for a beach to fully recover between storms is pivotal to long-term stability [72,73]. This study verified that with sufficient time between storms, the shoreline could return to a pre-storm state. Additionally, the balance between the net changes of the retreat and advance phases induced by the annual fluctuation is crucial to stability over time. A null balance would indicate that this process does not affect the shoreline dynamics on greater timescales than a year.

Any deviation from this equilibrium may lead to long-term shoreline advance or retreat. As the site exhibited a trend toward shoreline retreat in the historical dataset (−0.13 m per year), this suggests an imbalance in short-term processes or possibly the effect of an external influence, for example, human-induced factors such as sand mining, which the site has been exposed to during the second half of the 20th century [36].

Through a synthesis of observations, a conceptual model of the beach shoreline evolution has been formulated (see Figure 11). This model delineates both shoreline markers, highlighting the impacts of major storm events and less intense storm events (Figure 11b,c), along with the annual variations observed in the berm shoreline (as depicted in Figure 11b). Furthermore, the relationship between the changes in the berm shoreline and the $\eta 0$ is represented (Figure 11a), allowing for a clear visualization of the phases of erosion and accretion.

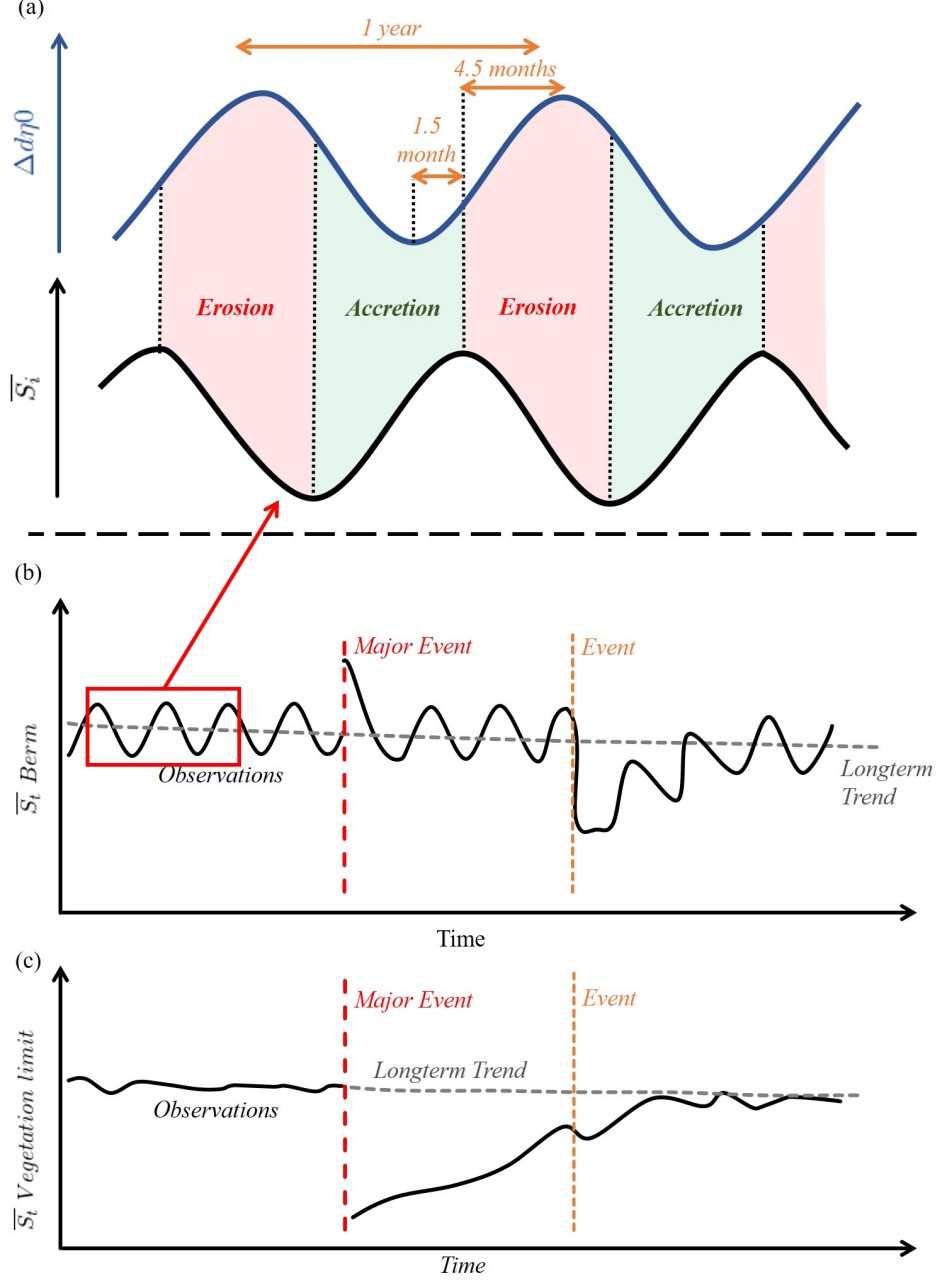

**Figure 11.** Conceptual models of berm shoreline (**a**) seasonal change and (**b**) longterm evolution. (**c**) Vegetation limit longterm change.

## 5. Future Expectations and Potential Solutions

Anticipating and mitigating future changes is crucial, especially considering the potential impacts of SLR and changes in storm intensity and frequency. The sea level rise (SLR) is a major consequence of climate change that will affect oceans globally [74]. By influencing the depths over the reef, SLR will diminish the filtering effect that coral reefs have on waves [75–77]. This could lead to a prolonged retreat phase in the annual shoreline cycle, resulting in an asymmetric pattern that may generate a continuous erosion trend. It has been evidenced that a healthy coral reef's growth rate could counter SLR [78,79], but the current state of most Caribbean reefs [80–82] prevents this possibility. Nevertheless, recent studies focusing on coral reef restoration as protection against flooding have demonstrated that this solution is relevant and efficient [83,84] in combating SLR.

Moreover, several studies indicate a possible change in future cyclonic activity, with a tendency to increase in both the frequency and intensity of cyclonic events [85–90]. Even though the trend in cyclonic activity is still a subject of discussion, this projection could have serious implications for all coasts impacted by such events. More frequent and intense storms will likely result in less time for recovery between storms, leading to accumulated erosion over time.

A subsequent conceptual model evaluating the transformations associated with climate change is introduced (Figure 12). This model primarily illustrates the impact of SLR on the annual berm shoreline cycle, specifically how it narrows the accretion window and expands the erosion window. Additionally, it highlights the effect of recurrent storm events occurring with insufficient recovery periods. Both these processes contribute to an accelerated trend of erosion, marking a significant divergence from the current-day dynamic.

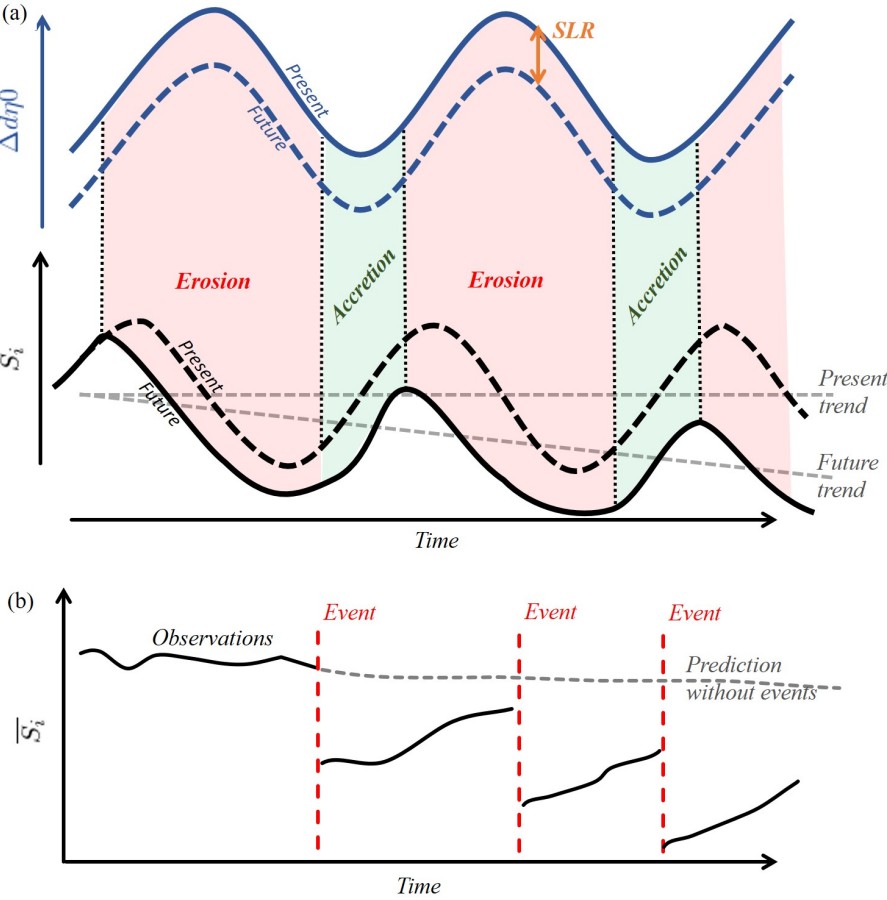

**Figure 12.** Conceptual models of future shoreline change taking into account (**a**) the effect of SLR on shoreline seasonality and (**b**) the longterm effect of the increase in cyclones frequency.

## 6. Conclusions and Perspectives

Through the integration of long-term, low-frequency (75 years) and short-term, high-frequency (3 years) datasets on shoreline dynamics, this study has elucidated the various processes at play across different timescales on a typical reef-lined Caribbean beach.

- Storms have a limited impact on long-term shoreline change, as sufficient recovery time allows a return to the pre-storm state. Extreme storm events have a significant influence on the vegetation limit shoreline, causing a retreat that may exceed 20 m and take several decades to recover, but that have little impact on the berm shoreline. More frequent and less intense storms, while having little influence on the vegetation limit, may strongly influence the berm shoreline.

- The annual cyclicity of $\eta 0$ is the driver of shoreline fluctuation, which is a novel observation. The wave attenuation by reefs is depth-dependent; $\eta 0$ seasonality triggers changes in the efficacy of the filter over the year, leading to significant annual fluctuation of the shoreline.

- Climate change, specifically SLR and potential changes in storm intensity and frequency, will affect shoreline dynamics. SLR may increase the depth over the reef, extending the retreat period and shortening the accretion phase within the $\eta 0$ cycle. Additionally, more frequent and intense storms could reduce the recovery time between events, possibly culminating in a future trend of increased shoreline retreat.

The findings of this study provide new insights into the complex dynamics of coastal processes. Future research must explore the underlying relationship between $\eta 0$ and shoreline change, quantify the morphological thresholds of storms, and identify the parameters that differentiate extreme storms and their impact on both advance/retreat and erosion/accretion.

Recent studies have proved that ecosystems are essential in hydrodynamic dampening, and thus in protecting the coastline. The effects of ecosystems such as coral reefs or upper beach vegetation on erosion and shoreline retreat are still fields of work but could be relevant solutions to address the challenges faced by our coastal environments when it comes to climate change.

**Author Contributions:** Conceptualization, T.L. and Y.B.; methodology, T.L.; software, T.L.; validation, T.L.; formal analysis, T.L. and Y.B.; investigation, T.L.; writing—original draft preparation, T.L.; writing—review and editing, T.L., Y.B., D.V.-L. and Y.D.L.T.; visualization, T.L.; supervision, T.L., Y.B., D.V.-L. and Y.D.L.T. project administration, Y.B. and Y.D.L.T.; funding acquisition, Y.B. and Y.D.L.T. All authors have read and agreed to the published version of the manuscript.

**Funding:** This research was funded by the EU INTEREG Caribbean CARIB-COAST project (grant number 4907).

**Institutional Review Board Statement:** Not applicable.

**Informed Consent Statement:** Not applicable.

**Data Availability Statement:** Data available on request.

**Acknowledgments:** The authors would like to thank M. Norden, T. Delahaye, and M. Moisan for their valuable help during the field experiments.

**Conflicts of Interest:** The authors declare no conflict of interest.

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
