# Peer review of "Seasonal to Multi-Decadal Shoreline Change on a Reef-Fringed Beach"

_2673-964X, doi:10.3390/coasts3030015_

Round 1
Reviewer 1 Report
The manuscript would need small changes before being published (see attached file). However, some questions need to be solved.
The authors consider “5 m” as a “reasonable margin of error” in the shoreline extraction process (lines 157-158). They should explain what evidence this estimate is based on.
Cyclone Fiona precedes the BRGM image dated 21/10/2022 in Table 1. Why was it not considered in the Multi Decadal Shoreline Observation?
The strong Cyclone Hugo (17/10/1989) immediately follows the IGN image dated 14/10/1989. The next image is almost 10 years more recent (01/02/1999) as reported in Table 1. Can this long gap influence the analysis you made?
However, the authors wrote “Hurricane Hugo passed … A satellite image taken one month after the hurricane showed obvious signs of damage, such as widespread sedimentary deposits and destruction of vegetation. The vegetation limit retreated by 25 m, while the berm shoreline advanced by a 5 m compared to the last shoreline” (lines 348-352). What "satellite image taken one month after the hurricane" are the authors referring to? In Table 2, after image Id 13, the next one is dated 01/02/1999.

Author Response
Dear Reviewer,
Firstly, I would like present my sincere thanks to you for the time and effort you have invested in reviewing this paper.
I want you to know that I have addressed all the comments that you highlighted, and I have marked each modification I made in red in the manuscript for easy identification. Please note that this includes changes made based on suggestions from other reviewers as well. I attached a new version of the manuscript to this reply.
Additionally, I have completely rewritten the discussion section, introducing a new workflow to enhance its readability and make it easier to follow. It's important to note that the discussion is not marked in red, though.
Below, you will find my response to each of your specific comments:
Reviewer 1: The authors consider “5 m” as a “reasonable margin of error” in the shoreline extraction process (lines 157-158). They should explain what evidence this estimate is based on.
- Answer:
- You are right. This value was chosen based on what was selected in the literature for similar resolution images. For example, in Doherty et al., 2019, they estimated their shoreline detection accuracy to be 3.5 m with images with a resolution of 2 m. Another example is Garcia-Rubio et al., 2015, who calculated a 5.6 m accuracy using 10 m resolution images and validation from on-site shoreline measurements.
- I chose the term "reasonable" as the margin is maximizing. It was taken using the image with the worst resolution (2m) to take into account the variable quality of the images (sun and shades, black and white images).
- I have added these elements to the text.
Reviewer 1: Cyclone Fiona precedes the BRGM image dated 21/10/2022 in Table 1. Why was it not considered in the Multi Decadal Shoreline Observation?
- Answer:
- Fiona was not considered in the multi-decadal analysis because it did not exceed the intensity threshold of sustained wind speed that was applied to this dataset. Morevover, as seen on the short-term evolution analysis, the impacts of this event were very limited on the site, generating a small advance in the middle of beach and a small retreat at its extremities.
Reviewer 1: The strong Cyclone Hugo (17/10/1989) immediately follows the IGN image dated 14/10/1989. The next image is almost 10 years more recent (01/02/1999) as reported in Table 1. Can this long gap influence the analysis you made?
- Answer:
- I am truly sorry; this misunderstanding is the result of a typo. Hurricane Hugo hit Guadeloupe on the 17th of September 1989 (17/09/1989) and not October.
- Additionally, the image taken a month after the event was not from a satellite but an aircraft. I also changed the text accordingly.
Best regards,
Thibault Laigre for the authors

Reviewer 2 Report
This is an interesting article that investigates shoreline dynamics of a Caribbean beach by means of a long-term satellite dataset spanning 75 years and a short data set of 3 years. The findings show that storms affect shoreline dynamics and the retreat of vegetation limit, and recovery may take decades. Short-term shoreline dynamics is driven by annual changes in water level. These patterns may change in the context of climate change.
I think that the idea is novel and highly relevant, given increasing occurrence of tropical cyclones worldwide.
The study is well done, but the discussion and conclusions could be improved, since they basically summarize the results. I have made several detailed comments bellow, so that the ms. can be improved. Also, some loose ends could be improved.
Based on the above, because the study is interesting and the results are good, my recommendation is to accept after major revision.
Specific comments follow:
1. Introduction
The information in the introduction is very interesting and clearly stated. The information is sufficient and clear. Similarly, the goals of the study were clearly explained.
2. Study area and methodology
2.1 Study site
Why is the beach under study been subject to consistent erosion from 1950 to 2013? Do goats eat Ipomoea pes-caprae? Does this herbivory affect beach erosion? How many goats are there grazing on the beach? This information is relevant to understand shoreline dynamics. Also, were there only three plant species at the beach?
2.2 Hydrodynamic conditions
2.2.1 Waves and still water level.
Well-explained.
2.2.2. Cyclones tracks
Well explained.
2.3. Shoreline datasets
Well explained
2.4. Data analyses
Well explained
3. Results
Interesting. Not easy to follow for non-experts in numerical modelling, but the results seem conclusive.
Discussion
The authors mention that the overall long-term evolution of the shoreline is retreating. The explanation of this long-time trend seemed a bit inconclusive. Major events seem to exert a low influence in shoreline retreat, by seasonal variation may be more relevant for long-term evolution of the beach.
The discussion was difficult to follow, because it mostly summarized or repeated the results. Perhaps the authors could focus on short- and long-term trends and explain the key drivers?
Also, what are the caveats of the study? How could it be improved?
The concluding remarks stating that coral reefs and coastal dunes may be used as ecosystem-based solutions (lines 555-566). Although I agree with this idea, the study does not provide evidence in this regard. So theses conclusions do not derive directly from the study.
5. Conclusions and perspectives
They seem more like a summary of the study than a concluding section. I recommend the authors to rewrite this section and focus on the take-home message.
Author Response
Dear Reviewer,
Firstly, I would like to extend my sincere thanks for the time and effort you have invested in reviewing this paper.
I have addressed all the points that you highlighted, and I have marked each modification in red within the attached manuscript for easy reference. This includes changes made based on suggestions from other reviewers.
Additionally, I have completely rewritten the discussion section, introducing a new workflow to enhance its readability and make it easier to follow. Please note that these changes to the discussion section are not marked in red.
Below, you will find my detailed response to each of your specific comments:
Reviewer 2: Why has the beach under study been subject to consistent erosion from 1950 to 2013?
- Answer:
- The main potential causes of erosion from 1950 to 2013, observed by Guillen et al., 2017, could be related to sand mining (a practice widespread in Guadeloupe during the second half of the century), coral reef degradation, and the concomitant increase in wave energy reaching the shoreline level, as mentioned by Bellwood et al., 2019.
In their study, Guillen et al.2017 used the vegetation limit as indicator. This vegetation limit is also highly dependent on human (trampling) and grazing activities…
Reviewer 2: Do goats eat Ipomoea pes-caprae? Does this herbivory affect beach erosion? How many goats are there grazing on the beach? This information is relevant to understand shoreline dynamics.
- Answer:
- I have no information stating that goats eat Ipomoea pes-caprae. However, they do eat the seedlings of the trees, hampering the growth of a new generation.
- The exact number of goats is hard to estimate as they are somewhat wild. I would say between 10 and 20 are regularly present around the site.
- I have no information regarding the direct impact of goats on erosion. They facilitate soil erosion in an indirect way by eating tree seedlings, thereby hampering the growth of new trees. When the "adult" trees die, the soil is left unprotected.
Reviewer 2: Also, were there only three plant species at the beach?
- Answer: No, there are other species, but I chose to cite only the principal ones for each category (i.e., crawling, layer, and shrubs). I changed the text to make this clearer.
Reviewer 2: The authors mention that the overall long-term evolution of the shoreline is retreating. The explanation of this long-time trend seemed a bit inconclusive. Major events seem to exert a low influence on shoreline retreat, but seasonal variation may be more relevant for the long-term evolution of the beach.
- Answer: We have reformulated the text for clarification. Stating that the erosion trend observed in the long term could be related to a residual effect of either storm recovery or annual cycles or the effect of another driver (possibly sand mining).
Reviewer 2: The discussion was difficult to follow because it mostly summarized or repeated the results. Perhaps the authors could focus on short- and long-term trends and explain the key drivers?
- Answer: We wrote a new version of the discussion following this advice
Reviewer 2: Also, what are the caveats of the study? How could it be improved?
- Answer: We added some elements about that in the discussion. Namely:
- the study was limited by the time resolution of the dataset. However, new technologies like satellite images and camera monitoring now allow to record data at a better resolution at a better frequency. Consequently, future observation will provide a better resolution around the passage of events, allowing further interpretation and quantification on the impact on the vegetation limit and berm shoreline as well as on recovery time.
- Additionally, we have added to the text that the camera-derived dataset could be complemented with direct measurements such as regular beach profiles measurement with a GNSS-RTK device. It would be useful to quantify the volume of sediment transfer and to evaluate if the change occurs at constant volume or if a portion of sediment is lost, which could explain a long-term erosion trend.

Round 2
Reviewer 1 Report
The authors have modified the text based on my comments and improved the first version of the manuscript.
Reviewer 2 Report
I think the authors made a good effort in modifying their very interesting article. I appreciate the effort they put into this.